# A multiscale geographically weighted regression analysis of teenage pregnancy and associated factors among adolescents aged 15 to 19 in Ethiopia using the 2019 mini-demographic and health survey

Tsion Mulat Tebeje[1]*, Mesfin Abebe[2], Fantu Mamo Aragaw[3], Beminate Lemma Seifu[4], Kusse Urmale Mare[5], Ever Siyoum Shewarega[6], Gizaw Sisay[1], Binyam Tariku Seboka[1]

1 School of Public Health, College of Health Science and Medicine, Dilla University, Dilla, Ethiopia, 2 Department of Midwifery, College of Health Science and Medicine, Dilla University, Dilla, Ethiopia, 3 Department of Epidemiology and Biostatistics, Institute of Public Health, College of Medicine and Health Sciences, University of Gondar, Gondar, Ethiopia, 4 Department of Public Health, College of Medicine and Health Sciences, Samara University, Afar, Ethiopia, 5 Department of Nursing, College of Medicine and Health Sciences, Samara University, Afar, Ethiopia, 6 Department of Reproductive Health, School of Public Health, College of Health Science and Medicine, Dilla University, Dilla, Ethiopia

* yemarina12@gmail.com

## Abstract

### Background

Teenage pregnancy remains one of the major reproductive health problems, especially in sub-Saharan African countries. It can lead to maternal and neonatal complications and social consequences. The proportion of teenage pregnancy differs across regions of Ethiopia. Thus, this study aimed to determine the spatial variation in determinants of teenage pregnancy among adolescents aged 15–19 years in Ethiopia using the 2019 Demographic and Health Survey (DHS).

### Methods

This study included a total weighted sample of 2165 teenage girls aged 15 to 19 years. A mixed-effect binary logistic regression model was employed to consider the hierarchical nature of the DHS data using STATA version 17. Adjusted odds ratios with 95% confidence intervals are reported, and a p-value less than 0.05 was used to identify significant predictors. The spatial analysis was conducted with ArcGIS version 10.7 and Python 3. To identify factors associated with the hotspots of teenage pregnancy, a multiscale geographically weighted regression (MGWR) was performed. Spatial regression models were compared using adjusted R2, the corrected Akaike information criterion (AICc), and the residual sum of squares (RSS).

**Data Availability Statement:** The data used in this research are publicly available from the measure

DHS program via an online request at https://dhsprogram.com/data/dataset_admin/login_main.cfm?CFID=10818526&CFTOKEN=c131014a480fe56-4E0C6B7F-F551-E6B2-50.

**Funding:** The author(s) received no specific funding for this work.

**Competing interests:** The authors have declared that no competing interests exist.

**Abbreviations:** AICc, corrected Akaike information criterion; AOR, adjusted odds ratio; CI, Confidence interval; CSA, central statistical agency; DHS, Demographic and Health Survey; EAs, Enumeration areas; EDHS, Ethiopian Demographic and Health Survey; GWR, Geographically weighted regression; HHH, Household Head; ICC, Intraclass correlation coefficient; LMICs, low- and middle-income countries; MOR, median odds ratio; MGWR, Multiscale Geographically Weighted Regression; PCV, Proportional change in variance; RSS, Residual sum of squares; OLS, Ordinary least squares; SSA, Sub-Saharan Africa; SNNPR, Southern Nations Nationalities and Peoples Region.

## Results

The prevalence of teenage pregnancy among adolescents aged 15 to 19 years was 12.98% (95% CI: 11.6%, 14.5%). It was spatially clustered throughout the country with a significant Moran's I value. Significant hotspot areas were detected in central and southern Afar; northern, central, and western Gambela; northeastern and southern central Oromia; and the eastern Somali region. The MGWR analysis revealed that the significant predictors of spatial variations in teenage pregnancy were being illiterate and being married. Based on the multivariable multilevel analysis, age 17 (AOR = 3.54; 95% CI: 1.60, 7.81), 18 (AOR = 8.21; 95% CI: 3.96, 17.0), 19 (AOR = 15.0; 95% CI: 6.84, 32.9), being literate (AOR = 0.57; 95% CI: 0.35, 0.92), being married (AOR = 22.8; 95% CI: 14.1, 37.0), age of household head (AOR = 0.98; 95% CI: 0.98, 0.99) and residing in the Gambela region (AOR = 3.27; 95% CI: 1.21, 8.86) were significantly associated with teenage pregnancy among adolescents aged 15 to 19.

## Conclusion

Teenage pregnancy is a public health problem in Ethiopia. Policymakers should prioritize addressing early marriage and improving teenage literacy rates, with a focus on the Gambela region and other hotspot areas. It is crucial to implement policies aimed at transforming the traditional practice of early marriage and to take measures to enhance literacy levels and promote awareness about sexual and reproductive health at the family and school levels. This will help ensure that young people have the opportunity to pursue education and make informed decisions about their reproductive health.

## Introduction

The highest number of teenage childbirths occurs in developing countries, where many girls marry early in life due to poverty, limited educational opportunities, cultural and social norms, and weak enforcement against child marriage [1–3]. Approximately 20 countries in sub-Saharan Africa (SSA) and Asia are responsible for 82% of adolescent maternal mortality [4]. This high rate could have resulted from the lower utilization of maternal health services by adolescents, coupled with their physical immaturity and discrimination [5,6]. Girls aged 15 to 19 years have a 28% greater risk of maternal mortality compared to women aged 20 to 24 years, influenced by physiological and bio demographic factors [7,8]. Pregnancy-related conditions are a significant cause of death among adolescent girls aged 15–19 years worldwide [9]. Girls aged 15 to 19 years make up 11% of all births worldwide [10]. Although there has been a decline in the rate of adolescent births, the total number of births has not decreased as much due to the growing adolescent population [11]. Each year, adolescents from low- and middle-income countries experience an estimated 21 million pregnancies, resulting in 12 million births [12]. Half of these pregnancies are unintended, and 55% of the unintended pregnancies often end in unsafe abortions [13]. Annually, 3.2 million unsafe abortions take place among adolescents aged 15 to 19, accounting for 15% of the total global incidence [14].

Adolescents have a significant unmet need for sexual and reproductive health care. A total of 43% of girls aged 15 to 19 years have an unmet need for modern contraception, which is higher than that of women of reproductive age, who have a 24% unmet need [12]. Adolescent

pregnancies lead to health consequences such as increased risks of eclampsia, puerperal endometritis, and systemic infections. There are also social consequences, including stigma, rejection, violence by partners or parents, and dropping out of school, which are commonly observed [15]. Additionally, adolescent pregnancy is related to lower child survival, which is attributable to biological and social risks [16].

Teenage pregnancy is a significant public health issue in Ethiopia, where there is limited knowledge and access to reproductive health services, restricted educational opportunities, and unwanted marriage at a young age [17]. The country ranks third in terms of adolescent maternal deaths [4]. Data from the 2000, 2005, 2011, and 2016 Ethiopian Demographic and Health Survey (EDHS) show that teenage pregnancy rates in Ethiopia were 16%, 17%, 12%, and 13%, respectively [18–21]. This overall decline demonstrates how public health interventions have effectively altered societal perceptions of early marriage and provided adolescents with access to reproductive health services [22]. However, the inconsistent trends indicate the need for continuous efforts. When broken down by region, the highest percentage was in the Afar region (23%), and the lowest percentage was in the capital, Addis Ababa (3%) [21].

According to previous research, factors related to adolescent pregnancy in LMICs include lower education and socioeconomic status, child marriage, not utilizing contraception, child sexual abuse and early sexual activity, age, religious beliefs, limited knowledge of reproductive and sexual health, gender inequality, physical or sexual violence, lower family income, being in the 18–19 age group, maternal history of teenage pregnancy, and inadequate communication with parents about reproductive health issues [23–28].

Fertility and childbearing among young people are among the most neglected issues in Ethiopia [29]. Ethiopian efforts for adolescents and youth are dispersed across ministries, lack coordination and meaningful youth involvement, have insufficient funding, focus on specific projects, and ineffectively implement policies [30]. Some similar studies have been conducted in the country [31–35], but they were based on the 2016 EDHS. Four of the aforementioned studies [31–34] investigated the trend and determinants of teenage pregnancy and did not take into account the geographical variation in teenage pregnancy in Ethiopia. However, one of them [35] examined regional disparities and studied the factors that influence regional disparities in teenage pregnancy using geographically weighted regression (GWR). Our study used the most recent data from the mini-EDHS 2019 to provide updated information. Additionally, this study was conducted to address the limitation of the previous study, which used GWR to assess the relationship between teenage pregnancy and its determinants. GWR assumes that the relationship between the outcome and its predictors operates at the same spatial scale. This disregards the fact that different relationships may occur at various scales due to heterogeneity or nonstationarity in geospatial relationships. To address this methodological gap, we used Multiscale Geographically Weighted Regression (MGWR), which considers multiple scales and relaxes the assumption of GWR by determining parameter-specific bandwidths (the spatial scale at which the underlying spatial processes operate). This allows the relationships between independent and dependent variables to vary at different spatial scales [36–38]. The purpose of this study was to better understand the spatial context of the determinants of teenage pregnancy using an explicit multiscale approach. Therefore, our study aimed to explore the spatial determinants of teenage pregnancy among adolescents aged 15 to 19 years using the most recent EDHS.

## Materials and methods

The 2019 mini-EDHS was conducted in Ethiopia through a community-based cross-sectional study. After Nigeria, Ethiopia is the second most populous country in Africa, with a total

population of almost 126.5 million people in 2023 and an area of 1.1 million square kilometers. Children under the age of 15, individuals aged 15 to 64, and those aged 65 or above represent 44%, 52%, and 4% of the total population, respectively [39,40]. The country is subdivided into nine regions and two city administrations.

The 2019 mini-EDHS sample was stratified and selected in two stages. Each region was stratified into urban and rural areas, yielding 21 sampling strata. Samples of enumeration areas (EAs) were selected independently in each stratum in the two stages. The first stage involved a selection of 305 EAs, with 93 in urban areas and 212 in rural areas. In the second stage, a fixed number of 30 households per cluster was selected with an equal probability of systematic selection from the newly created household listing [39]. We utilized the individual data (IR) set from the mini-EDHS 2019 for the analysis, and we obtained longitude and latitude coordinates at the cluster or EA level. The Ethiopian administrative boundaries shapefile was obtained from the Central Statistical Agency (CSA).

The outcome variable is teenage pregnancy, defined as women aged 15–19 years who had a live birth, who were pregnant with their first child, or who had begun childbearing at the time of the survey [41]. It is dichotomized as 0 = no (those who did not experience teenage pregnancy) and 1 = yes (those who experienced teenage pregnancy). The percentage of women who had begun childbearing was calculated by dividing the number of women who either had a birth or who were pregnant at the time of the interview by all women aged 15–19 years [41]. The independent variables included were age, marital status, religion, sex of the household head, education level, wealth index, media exposure, and contraceptive utilization. The community-level variables were residence and region; we also created aggregate community-level variables, which were generated by aggregating individual-level factors at the cluster level: community education, community poverty, community contraceptive utilization, community media exposure, and community percentage of marriage.

## Data processing and analysis

**Factors associated with teenage pregnancy.** Due to the hierarchical nature of the EDHS data, teenagers within the same cluster exhibit similar characteristics compared to those in another cluster. Therefore, a multilevel binary logistic regression model that accounts for the heterogeneity between clusters was constructed. To assess the clustering effect, the intraclass correlation coefficient (ICC) was estimated. The ICC showed a significant clustering effect (ICC = 20.9%). The median odds ratio (MOR), which is the median value of the odds ratio between the highest risk area and lowest risk area when randomly picking out two areas, is the increased risk that would exist when moving to a different area with a higher level of risk [42]. The proportional change in variance (PCV) is the variation in the dependent variable explained by all individual- and community-level factors [43].

Four models were constructed for the multilevel logistic regression analysis. The first model was a null model without explanatory variables to determine the degree of cluster variation in teenage pregnancy. The second model was fitted with individual-level variables, the third model was fitted with community-level variables, and the fourth model was fitted with both individual- and community-level variables at the same time. Variables with p values < 0.25 in the bi-variable analysis were added to the multivariable model. The associations between the dependent and independent variables were assessed and are presented as adjusted odds ratios (AORs) with 95% confidence intervals (CIs) at p values < 0.05.

**Spatial analysis.** A spatial analysis was conducted by using ArcGIS version 10.7 and Python 3. To assess whether the spatial distribution of teenage pregnancy in Ethiopia is dispersed, clustered, or randomly distributed, global spatial autocorrelation (global Moran's I)

was utilized. A Moran's I value < 0.05 indicated that the distribution of teenage pregnancy in Ethiopia was nonrandom [44]. Hotspot analysis was conducted to identify statistically significant clustering areas using Getis-ordGi* statistics, which resulted in z scores and p-values indicating where high and low values of teenage pregnancy clustered spatially [45].

To comprehend the association between the density of a certain event and other various sociodemographic and environmental aspects of the population, spatial regression models are important [46]. After the identification of the high-risk areas (hotspot areas) of teenage pregnancy, factors that contributed to the spatial clustering of teenage pregnancy in these areas were determined at the EA level by a global and local spatial regression model: ordinary least squares (OLS) and GWR and MGWR, respectively. The spatial nonstationarity of the proportion of teenage pregnancy was ascertained by using global spatial autocorrelation before fitting the OLS, GWR, and MGWR models.

First, OLS was fitted to choose variables that are appropriate for the spatial variation of teenage pregnancy. Before proceeding to the GWR model, we first checked the six assumptions of the OLS model using exploratory regression. Exploratory regression is a data mining tool that helps us determine which combinations of explanatory variables meet all necessary OLS diagnostics [47]. We checked for the expected relationship of the explanatory variables, the statistical significance of each explanatory variable, the randomness of residuals (which should be normally distributed with no spatial patterns), the statistical nonsignificance of the Jarque-Bera statistics, freedom from multicollinearity in the explanatory variables (VIF < 7.5), and the model performance (strength of the adjusted R-squared) [48]. The Koenker Breusch–Pagan (BP) test was significant, indicating nonstationarity or heterogeneity in the relationship between the dependent and independent variables. As a result, we utilized local spatial regression models (GWR and MGWR), which account for variations in the relationships between variables across different spatial locations, for this study [49].

A geographically weighted regression model was used to determine the predictor variables for teenage pregnancy among adolescents aged 15 to 19 years. The proportion of the outcome variable and all predictors was calculated for each cluster. Classical GWR assumes that all of the processes being modeled operate at the same spatial scale. However, the MGWR reduces this assumption by allowing different processes to operate at different spatial scales by deriving an optimal bandwidth vector in which each element indicates the spatial scale at which a particular process takes place. Hence, MGWR is preferable because covariate-specific bandwidths are obtained rather than a single average bandwidth and provides valuable information on the scale at which different processes operate. Model calibration and bandwidth vector selection in MGWR were conducted using a back-fitting algorithm. Model fitness was compared among the global and local models using the Akaike information criterion (AIC), residual sum of squares (RSS), and adjusted R-squared values [50,51].

## Ethical approval and consent to participate

The data were obtained from the Demographic and Health Surveys (DHS) Program and can be freely accessed from the program website (www.dhsprogram.com). As the study was a secondary data analysis of publicly available data from the MEASURE DHS program, ethical approval and participant consent were not necessary for this particular study. We requested the use of the DHS Program, and permission was granted to download and use the data for this study. Approval for the use of the data was obtained from the Measure DHS program, and the dataset was downloaded from https://dhsprogram.com/data/available-datasets.cfm. The IRB-approved procedures for DHS public-use datasets do not allow the identification of respondents, households, or sample communities. Geographic identifiers only go down to

regional levels, and each EA has a primary sampling unit (PSU) number without labels. Surveys collect GIS coordinates for the entire enumeration area, preventing specific enumeration areas from being identified.

## Results

### Descriptive results

A total of 2165 teenage girls aged 15 to 19 years were included in the study. The median age of the respondents was 17 years. One-fifth (20.4%) of the participants were married, and only 9.5% of the girls utilized contraceptives. One-third of the girls were urban residents, while a quarter of them (25.1%) were illiterate. Regarding regional states, the majority of the participants were from the Oromia region (851, 39.3%), and the minority of the participants were from the Harari region (6, 0.29%) (Table 1).

**Prevalence of teenage pregnancy in Ethiopia.** The prevalence of teenage pregnancy among adolescents aged 15 to 19 years was 12.98% (95% CI: 11.63%, 14.46%). The highest percentage of teenage pregnancy was observed in the Afar region (28.3%), followed by the Gambella region (27.4%). The lowest prevalence was observed in Addis Ababa (3.8%) (Table 1).

### Spatial analysis

Global spatial autocorrelation analysis revealed that the spatial distribution of teenage pregnancy in Ethiopia significantly varied across the country, with a global Moran's index value of 0.04 and a p-value < 0.001. The Z score of 4.71 indicated that there was less than a 1% likelihood that the clustered pattern could be a result of random chance. (Fig 1).

Significant hotspot areas (areas with a high proportion of teenage pregnancy) of teenage pregnancy were observed in central and southern Afar; northern, central, and western Gambela; northeastern and southern central Oromia; and the eastern Somali region. Significant cold spot areas (areas with a low proportion of teenage pregnancy) were identified in central and southern Amhara, central Oromia, Addis Ababa, and northern SNNPR. (Fig 2).

**Spatial regression analysis.** To explore the assumptions of spatial regression and to estimate the coefficients of the selected variables on teenage pregnancy, OLS regression was conducted. The OLS model identified predictors of each hotspot of teenage pregnancy. The joint F-statistics and Wald statistics were significant, which shows that the model was statistically significant. The Koenker statistics was statistically significant, which is indicative of nonstationarity or heterogeneity of the relationship between the dependent and the independent variables across the study areas. Therefore, GWR was applied because it assumes that the relationship between the independent and dependent variables has spatial heterogeneity (as confirmed by Koenker statistics). No multicollinearity was observed between the selected explanatory variables (Table 2).

The local models, GWR and MGWR, were implemented to capture spatial heterogeneity. Both of the local models had better model fitness than the global model. The adjusted $R^2$ improved from 0.547 (OLS) to 0.634 (GWR) and 0.691 (MGWR). The adjusted $R^2$ indicates that the GWR and MGWR have 8.7% and 14.4% increased explanatory power, respectively, compared to the OLS. The residual sum of squares (RSS), which indicates unexplained variations, was very high in the OLS model (134.6) and was 93.7 (GWR) and 78.4 (MGWR). The local regression showed an AICc of 630.4, while the GWR reduced the AICc to 618.6, and the MGWR further decreased the AICc to 570.4. Relying on this information, it was confirmed that the local models improved the model performance (Table 3).

The GWR model has a single bandwidth of 103; thus, 103 nearest neighbors are considered to inform the construction of parameter estimates at each local regression point. This implies

**Table 1. Descriptive characteristics of the study participants with a prevalence of teenage pregnancy in Ethiopia, 2019 EDHS.**

| Variables | | Weighted frequency (percentage) | Teenage pregnancy | |
|---|---|---|---|---|
| | | | Yes (%) | No (%) |
| Total | | 2165 (100) | 281 (12.98) | 1884 (87.02) |
| Age | 15 | 506 (23.4) | 6 (1.07) | 500 (98.9) |
| | 16 | 451 (20.8) | 21 (4.67) | 430 (95.3) |
| | 17 | 328 (15.2) | 37 (11.4) | 291 (88.6) |
| | 18 | 608 (28.1) | 139 (23.0) | 468 (77.0) |
| | 19 | 272 (12.6) | 77 (28.5) | 195 (71.5) |
| Religion | Orthodox | 928 (42.9) | 86 (9.2) | 842 (90.8) |
| | Muslim | 589 (27.2) | 79 (13.5) | 510 (86.5) |
| | Protestant | 614 (28.4) | 114 (18.7) | 499 (81.3) |
| | Others | 34 (1.6) | 2 (4.2) | 33 (95.8) |
| Literacy | Illiterate | 543 (25.1) | 161 (29.7) | 382 (70.3) |
| | Literate | 1622 (74.9) | 120 (7.4) | 1502 (92.6) |
| Education level | No education | 229 (10.6) | 75 (32.7) | 154 (67.3) |
| | Primary | 1437 (66.4) | 177 (12.3) | 1259 (87.7) |
| | Secondary | 432 (19.9) | 19 (4.3) | 413 (95.7) |
| | Higher | 67 (3.1) | 10 (14.9) | 57 (85.1) |
| Marital status | Single | 1725 (79.7) | 50 (2.9) | 1675 (97.1) |
| | Married | 440 (20.4) | 231 (52.5) | 209 (47.5) |
| Contraceptive utilization | Yes | 206 (9.5) | 93 (45.3) | 113 (54.7) |
| | No | 1959 (90.5) | 188 (9.6) | 1772 (90.4) |
| Wealth index | Poor | 720 (33.2) | 124 (17.2) | 596 (82.8) |
| | Middle | 441 (20.4) | 71 (16.1) | 370 (83.9) |
| | Rich | 1004 (46.4) | 87 (8.6) | 917 (91.4) |
| Media exposure | No | 1361 (62.9) | 193 (14.2) | 1168 (85.8) |
| | Yes | 804 (37.1) | 88 (11.0) | 716 (89.0) |
| Sex of household head | Male | 1658 (76.6) | 240 (14.5) | 1418 (85.5) |
| | Female | 507 (23.4) | 41 (8.1) | 466 (91.9) |
| **Community-level variables** | | | | |
| Residence | Urban | 720 (33.2) | 71 (9.9) | 648 (90.1) |
| | Rural | 1445 (66.8) | 210 (14.5) | 1236 (85.5) |
| Region | Tigray | 138 (6.4) | 16 (11.6) | 122 (88.4) |
| | Afar | 16 (0.75) | 5 (28.3) | 11 (71.7) |
| | Amhara | 498 (23.0) | 35 (7.0) | 464 (93.0) |
| | Oromia | 851 (39.3) | 129 (15.2) | 722 (84.8) |
| | Somali | 111 (5.1) | 20 (17.6) | 91 (82.4) |
| | Benishangul | 25 (1.2) | 5 (17.8) | 20 (82.2) |
| | SNNPR | 402 (18.6) | 63 (15.8) | 339 (84.2) |
| | Gambella | 9 (0.42) | 2 (27.4) | 7 (72.6) |
| | Harari | 6 (0.29) | 1 (14.3) | 5 (85.7) |
| | Addis Ababa | 93 (4.3) | 4 (3.8) | 89 (96.2) |
| | Dire Dawa | 15 (0.7) | 2 (12.9) | 13 (87.1) |
| Community media exposure | Low proportion | 1115 (51.5) | 141 (12.7) | 974 (87.3) |
| | High proportion | 1050 (48.5) | 140 (13.3) | 910 (86.7) |
| Community education level | Low proportion | 869 (40.1) | 174 (20.0) | 695 (80.0) |
| | High proportion | 1296 (59.9) | 107 (8.3) | 1189 (91.7) |

(*Continued*)

**Table 1.** (Continued)

| Variables | | Weighted frequency (percentage) | Teenage pregnancy | |
|---|---|---|---|---|
| | | | Yes (%) | No (%) |
| Community wealth index | Low proportion | 991 (45.8) | 155 (15.6) | 836 (84.4) |
| | High proportion | 1174 (54.2) | 126 (10.7) | 1048 (89.3) |
| Community contraceptive use | Low proportion | 1142 (52.7) | 119 (10.4) | 1023 (89.6) |
| | High proportion | 1023 (47.3) | 163 (15.9) | 861 (84.1) |
| Community literacy | Low proportion | 1014 (46.8) | 203 (20.0) | 811 (80.0) |
| | High proportion | 1151 (53.2) | 78 (6.8) | 1073 (93.2) |
| Community marital status | Low proportion | 1083 (50.0) | 55 (5.1) | 1028 (94.9) |
| | High proportion | 1082 (50.0) | 226 (20.9) | 856 (79.1) |

that the GWR analysis focused on the 103 nearest neighboring data points to calculate parameter estimates. Using the 103 bandwidths for each variable, 63.4% of the observations were correctly classified. In the case of MGWR, the optimum bandwidth for each variable is computed. By allowing multiple bandwidths in MGWR, the model theoretically accounts for an optimal number of neighbors for each parameter estimate, thus allowing better predictions for the response variables. In MGWR, the bandwidths for each parameter are 111 for poor wealth index, 43 for being married, 301 for being Muslim, 301 for being illiterate, 292 for community poverty and 161 for community illiteracy. These specific bandwidths enable the model to adjust to the unique spatial characteristics of each variable (Table 4).

The figures below show the mapped GWR and MGWR coefficient estimates for the intercept and the covariates. In Fig 3, the local intercept in the MGWR is interpreted as the value of the outcome that would be expected if every location had exactly the same average value of

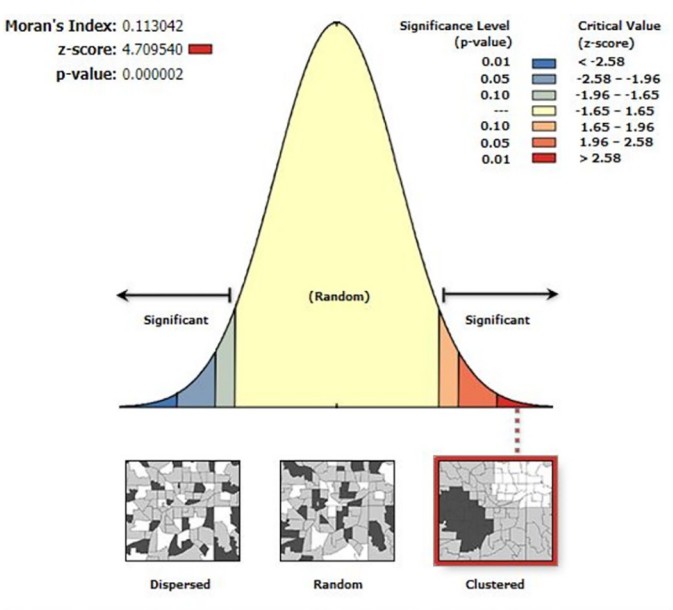

Given the z-score of 4.70954028255, there is a less than 1% likelihood that this clustered pattern could be the result of random chance.

**Fig 1. Global spatial autocorrelation analysis of teenage pregnancy among adolescents aged 15 to 19 years in Ethiopia, 2019.**

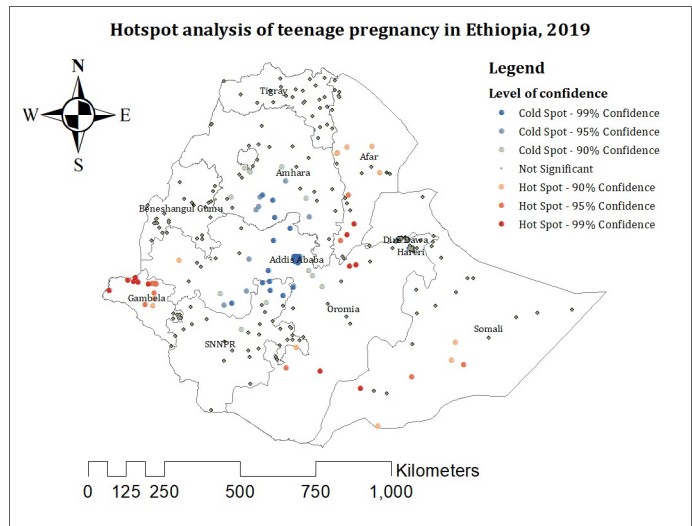

**Fig 2. The Getis Ord Gi statistical analysis of hotspots of teenage pregnancy among adolescents aged 15 to 19 years in Ethiopia, 2019.** Shapefile source: (CSA, 2013; https://africaopendata.org/dataset/ethiopia-shapefiles); Map output: Own analysis using ArcGIS Software.

each covariate. Based on the mapping of the intercept coefficients, the local intercept in the MGWR model can be interpreted as showing the level of teenage pregnancy that would be expected in each region if each of the regions had an average population profile or held all covariates constant. There is a high rate of teenage pregnancy in Gambella and some parts of western and northeastern Oromia. In contrast, there was a lower rate of teenage pregnancy in central and western Amhara. The intercept in the GWR was not significantly different from zero, but there was an outlier with a positive relationship in southern Afar at the border with Oromia.

The parameter estimates for the proportion of being married were positively associated with teenage pregnancy in Tigray, Amhara, Afar, Somali, eastern SNNPR, Benishangul,

**Table 2. The ordinary least squares analysis results for the 2019 EDHS.**

| Variable | Coefficient | Robust SE | Robust t-statistics | Robust probability | VIF |
|---|---|---|---|---|---|
| Intercept | 0.015 | 0.01 | 1.48 | 0.14 | - - - |
| Proportion of poor wealth index | 0.016 | 0.03 | 0.47 | 0.64 | 3.30 |
| Proportion of married teenagers | 0.58 | 0.05 | 10.85 | 0.000000* | 1.24 |
| Proportion of Muslim teenagers | -0.026 | 0.017 | -1.54 | 0.13 | 1.21 |
| Proportion of illiterate teenagers | 0.098 | 0.047 | 2.09 | 0.036839* | 2.95 |
| Proportion of community poverty | 0.025 | 0.02 | 1.05 | 0.29 | 2.96 |
| Proportion of community illiteracy | -0.040 | 0.02 | -1.77 | 0.079 | 2.51 |
| **Ordinary least square regression diagnostics** | | | | | |
| Number of observations | 303 | Adjusted R-squared | | | 0.547 |
| Joint F-statistics | 61.7 | Prob(>F), (6,296) degree of freedom | | | 0.000000* |
| Joint Wald statistics | 258.7 | Prob (> chi-squared), (6) degree of freedom | | | 0.000000* |
| Koenker (BP) statistics | 49.9 | Prob (> chi-squared), (6) degree of freedom | | | 0.000000* |
| Jarque–Bera | 60.1 | Prob (> chi-squared), (2) degree of freedom | | | 0.000000* |

*p value < 0.05.

**Table 3. Comparison of the goodness-of-fit measures between the global (OLS) and local (GWR and MGWR) models.**

| Model comparison parameter | Adj-R$^2$ | RSS | AICc |
|---|---|---|---|
| Model | | | |
| OLS | 0.547 | 134.573 | 630.444 |
| GWR | 0.634 | 93.709 | 618.603 |
| MGWR | 0.691 | 78.424 | 570.350 |

**Table 4. Summary of the GWR and MGWR model results for the predictors of teenage pregnancy in Ethiopia in the 2019 EDHS.**

| Variable | Mean | STD | Min | Median | Max | Bandwidth |
|---|---|---|---|---|---|---|
| **GWR model** | | | | | | |
| Intercept | -0.034 | 0.093 | -0.231 | -0.017 | 0.284 | 103 |
| Proportion of poor wealth index | 0.056 | 0.160 | -0.249 | 0.051 | 0.399 | 103 |
| Proportion of married teenagers | 0.670 | 0.137 | 0.302 | 0.700 | 1.012 | 103 |
| Proportion of Muslim teenagers | -0.071 | 0.074 | -0.259 | -0.062 | 0.120 | 103 |
| Proportion of illiterate teenagers | 0.135 | 0.106 | -0.305 | 0.115 | 0.388 | 103 |
| Proportion of community poverty | 0.037 | 0.178 | -0.328 | 0.077 | 0.348 | 103 |
| Proportion of community illiteracy | -0.060 | 0.149 | -0.330 | -0.043 | 0.381 | 103 |
| **MGWR model** | | | | | | |
| Intercept | -0.052 | 0.197 | -0.373 | -0.079 | 0.662 | 50 |
| Proportion of poor wealth index | -0.008 | 0.101 | -0.222 | -0.050 | 0.292 | 111 |
| Proportion of married teenagers | 0.640 | 0.250 | -0.087 | 0.678 | 1.128 | 43 |
| Proportion of Muslim teenagers | -0.067 | 0.014 | -0.094 | -0.065 | -0.050 | 301 |
| Proportion of illiterate teenagers | 0.187 | 0.007 | 0.175 | 0.190 | 0.200 | 301 |
| Proportion of community poverty | 0.061 | 0.019 | 0.025 | 0.058 | 0.096 | 292 |
| Proportion of community illiteracy | -0.051 | 0.054 | -0.141 | -0.065 | 0.064 | 161 |

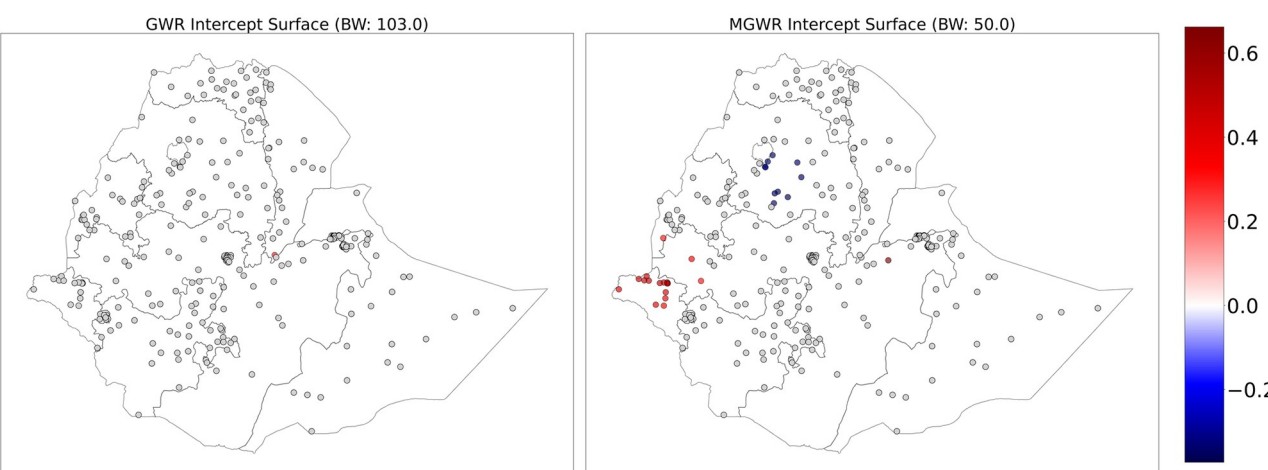

**Fig 3.** GWR (left) and MGWR (right) parameter estimates for the intercept showing local patterns of spatial heterogeneity. The gray dots are not significantly different from zero. Shapefile source: (CSA, 2013; https://africaopendata.org/dataset/ethiopia-shapefiles); Map output: Own analysis using Python Software.

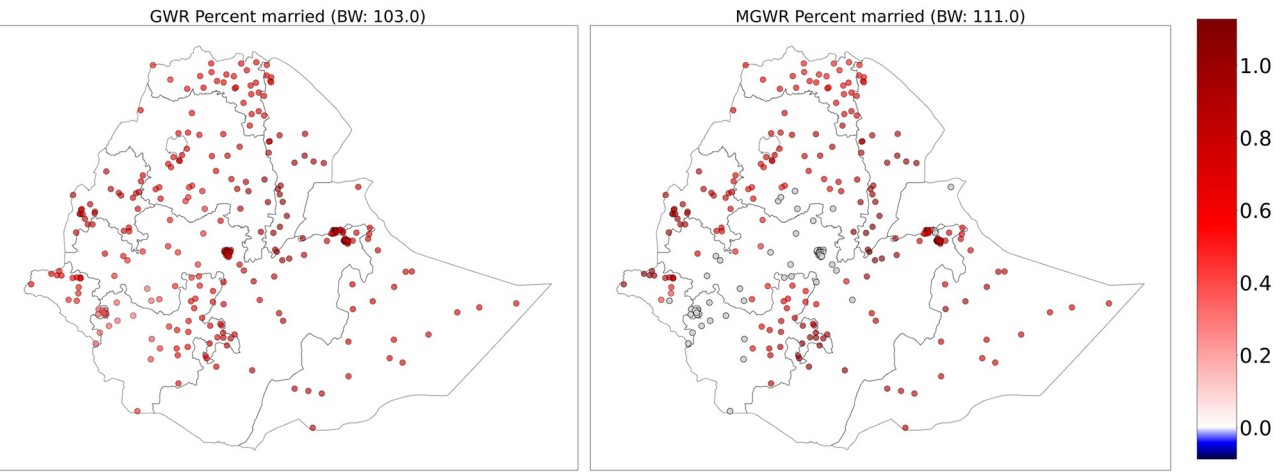

**Fig 4.** GWR (left) and MGWR (right) parameter estimates for the proportion of married women showing local patterns of spatial heterogeneity. The gray dots are not significantly different from zero. Shapefile source: (CSA, 2013; https://africaopendata.org/dataset/ethiopia-shapefiles); Map output: Own analysis using Python Software.

Gambela, Harari, Dire Dawa and eastern Oromia. The GWR model showed a positive relationship with the outcome variable across the study area with no spatial heterogeneity (Fig 4).

The proportion of being illiterate does not show any spatial heterogeneity, as it has a positive association with teenage pregnancy across the country. This implies that being illiterate is associated with teenage pregnancy throughout the country, with no spatial differences. In the GWR model, it was not significantly different from zero because it has no statistically nonzero parameter estimate (Fig 5).

## Multilevel analysis of factors associated with teenage pregnancy

Multilevel mixed-effect logistic regression analysis was conducted to identify individual- and community-level variables that were significantly associated with teenage pregnancy. The ICC

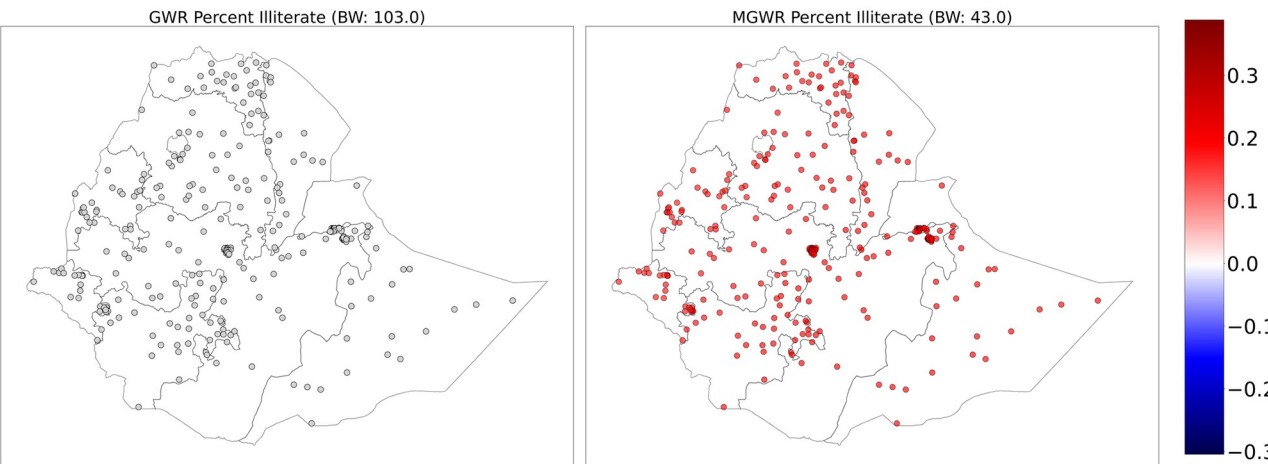

**Fig 5.** GWR (left) and MGWR (right) parameter estimates for the proportion of illiterate women showing local patterns of spatial heterogeneity. The gray dots are not significantly different from zero. Shapefile source: (CSA, 2013; https://africaopendata.org/dataset/ethiopia-shapefiles); Map output: Own analysis using Python Software.

**Table 5. Parameters and model fit statistics for multilevel models.**

| Parameters | Null model | Model 2 | Model 3 | Model 4 |
|---|---|---|---|---|
| Community level variance (SE) | 0.87 | 0.46 | 0.015 | 0.24 |
| ICC | 0.21 | 0.12 | 0.0046 | 0.069 |
| MOR | 2.43 | 1.91 | 1.125 | 1.589 |
| PCV | Ref | 0.47 | 0.98 | 0.72 |
| **Model fitness** | | | | |
| Deviance | 1715.1 | 946.82 | 1536.88 | 920.48 |
| Log-likelihood | -857.55 | -473.41 | -768.44 | -460.24 |

in the null model was 0.209, which shows that 20.9% of the variation in teenage pregnancy is explained by community-level variables (between-cluster variability). According to the MOR in the null model (2.43), there was a variation in teenage pregnancy between the clusters. When a teenage girl is switched from a low-risk cluster to a high-risk cluster, she is 2.43 times more likely to have a teenage pregnancy. According to the final model (model 4), the PCV showed that 72% of the total variability in teenage pregnancy was explained by both individual- and community-level factors. (Model 4) was the best-fitting model because it attained the highest log likelihood (in negative numbers) and the lowest deviance (Table 5).

Based on the findings of Model 4, the factors that were found to have a significant association with teenage pregnancy were the age of the girls, literacy status, marital status, age of the household head and residence in the Gambela region (Table 6).

The odds of having teenage pregnancy among girls aged 17, 18 and 19 years were 3.5 times greater (AOR = 3.54; 95% CI: 1.60, 7.81), 8.2 times greater (AOR = 8.21; 95% CI: 3.96, 17.0) and 15 times greater (AOR = 15.0; 95% CI: 6.84, 32.9), respectively, than teenagers aged 15 years. Literate teenagers had a 43% (AOR = 0.57; 95% CI: 0.35, 0.92) lower risk of teenage pregnancy than did those who were illiterate. Compared with single teenagers, married teenagers had 22.8 (AOR = 22.8; 95% CI: 14.1, 37.0) times greater odds of having a teenage pregnancy. With a one-year increase in the age of the household head, the odds of experiencing teenage pregnancy decreased by 2% (AOR = 0.98; 95% CI: 0.98, 0.99). Teenagers residing in the Gambela region had 3.27 (AOR = 3.27; 95% CI: 1.21, 8.86) times greater odds of having a teenage pregnancy than did those residing in the Tigray region. However, religion, education level, contraceptive use, wealth index, media exposure, and community variables, except region, were not statistically significant (Table 6).

## Discussion

The spatial statistics showed that the spatial distribution of teenage pregnancy among adolescents aged 15 to 19 years had significant variation in Ethiopia. The hotspot areas of teenage pregnancy were identified in central and southern Afar, northern, central and western Gambela, northeastern and southern central Oromia and the eastern Somali region. A possible explanation might be that most of these areas are pastoralists and semi pastoralists, characterized by seasonal mobility and lower utilization of contraceptives [52]. They also live in traditional settings and strongly adhere to traditional cultural values and beliefs, which leads to poor sexual and reproductive health outcomes and early marriage [53].

According to the spatial regression analysis of the global model, the significant predictors of teenage pregnancy hotspots among adolescents aged 15 to 19 years were marital status and illiteracy. Local models were applied to the same predictors used in OLS to explore the local spatial variation in the relationship with the proportion of teenage pregnancy. As the MGWR

**Table 6. Multilevel analysis of factors associated with teenage pregnancy among adolescents aged 15 to 19 years in Ethiopia, 2019 EDHS.**

| Explanatory variable | Null model | Model 2 AOR (95%CI) | Model 3 AOR (95%CI) | Model 4 AOR (95%CI) |
|---|---|---|---|---|
| **Age** | | | | |
| 15 | | 1 | | 1 |
| 16 | | 1.68 (0.70, 3.99) | | 1.77 (0.75, 4.18) |
| 17 | | 3.63 (1.64, 8.04) | | 3.54(1.60,7.81)** |
| 18 | | 8.38 (4.03, 17.4) | | 8.21(3.96,17.0)** |
| 19 | | 15.3 (7.02, 33.7) | | 15.0(6.84,32.9)** |
| **Religion** | | | | |
| Orthodox | | 1 | | 1 |
| Muslim | | 1.13 (0.69, 1.81) | | 0.89 (0.47, 1.68) |
| Protestant | | 2.14 (1.26, 3.64) | | 1.62 (0.85, 3.10) |
| Others | | 1.46 (0.35, 6.07) | | 1.08 (0.24, 4.83) |
| **Literacy** | | | | |
| Illiterate | | 1 | | 1 |
| Literate | | 0.65 (0.41, 1.02) | | 0.57 (0.35, 0.92)* |
| **Education level** | | | | |
| No education | | 1 | | 1 |
| Primary | | 0.77 (0.46, 1.29) | | 0.84 (0.48, 1.45) |
| Secondary | | 0.59 (0.28, 1.26) | | 0.71 (0.32, 1.54) |
| Higher | | 0.37 (0.11, 1.30) | | 0.39 (0.11, 1.46) |
| **Marital status** | | | | |
| Single | | 1 | | 1 |
| Married | | 24.03(15.3,37.6) | | 22.8(14.1,37.0)** |
| **Contraceptive utilization** | | | | |
| Yes | | 1 | | 1 |
| No | | 0.97 (0.58, 1.63) | | 1.04 (0.59, 1.81) |
| **Wealth index** | | | | |
| Poor | | 1 | | 1 |
| Middle | | 0.97 (0.57, 1.67) | | 1.11 (0.63, 1.93) |
| Rich | | 0.49 (0.29, 0.80) | | 0.59 (0.32, 1.11) |
| **Media exposure** | | | | |
| No | | 1 | | 1 |
| Yes | | 1.25 (0.80, 1.93) | | 1.19 (0.74, 1.91) |
| **Age of HHH** | | 0.98 (0.97, 0.99) | | 0.98 (0.98, 0.99)* |
| **Residence** | | | | |
| Urban | | | 1 | 1 |
| Rural | | | 1.03 (0.68, 1.58) | 0.92 (0.50, 1.68) |
| **Region** | | | | |
| Tigray | | | 1 | 1 |
| Afar | | | 1.31 (0.68, 2.51) | 1.31 (0.44, 3.92) |
| Amhara | | | 0.50 (0.25, 1.01) | 0.45 (0.17, 1.18) |
| Oromia | | | 1.07 (0.59, 1.95) | 1.03 (0.39, 2.70) |
| Somali | | | 1.16 (0.59, 2.28) | 1.04 (0.34, 3.26) |
| Benishangul | | | 1.36 (0.72, 2.59) | 1.34 (0.48, 3.71) |
| SNNPR | | | 1.04 (0.55, 1.95) | 0.77 (0.28, 2.13) |
| Gambella | | | 2.04 (1.09, 3.81) | 3.27 (1.21, 8.86)* |
| Harari | | | 1.42 (0.70, 2.86) | 1.26 (0.39, 4.07) |

*(Continued)*

**Table 6.** (Continued)

| Explanatory variable | Null model | Model 2 AOR (95%CI) | Model 3 AOR (95%CI) | Model 4 AOR (95%CI) |
|---|---|---|---|---|
| Addis Ababa | | | 0.80 (0.29, 2.25) | 0.86 (0.23, 3.21) |
| Dire Dawa | | | 1.02 (0.52, 1.99) | 1.19 (0.39, 3.57) |
| **Community media exposure** | | | | |
| Low proportion | | | 1 | 1 |
| High proportion | | | 0.89 (0.65, 1.23) | 0.93 (0.50, 1.68) |
| **Community education level** | | | | |
| Low proportion | | | 1 | 1 |
| High proportion | | | 0.81 (0.58, 1.12) | 0.81 (0.49, 1.32) |
| **Community wealth index** | | | | |
| Low proportion | | | 1 | 1 |
| High proportion | | | 0.77 (0.53, 1.12) | 0.83 (0.45, 1.54) |
| **Community contraceptive use** | | | | |
| Low proportion | | | 1 | 1 |
| High proportion | | | 1.25 (0.91, 1.71) | 1.00 (0.61, 1.63) |
| **Community literacy** | | | | |
| Low proportion | | | 1 | 1 |
| High proportion | | | 0.89 (0.63, 1.25) | 1.28 (0.76, 2.15) |
| **Community marital status** | | | | |
| Low proportion | | | 1 | 1 |
| High proportion | | | 4.84 (3.35, 7.00) | 1.34 (0.79, 2.25) |
| Intercept | 0.14 (0.11, 0.17) | 0.03(0.009, 0.08) | 0.07(0.03, 0.15) | 0.02(0.005, 0.12) |

*p value < 0.05;

** p value < 0.01;

AOR, adjusted odds ratio; HHH, household head.

model had the smallest AICc and RSS values and the highest adjusted $R^2$ value, it is a statistically preferable local model, which is in line with previous studies [54].

The MGWR analysis revealed a significant positive relationship between the proportion of being a married women and teenage pregnancy in central and southern Afar, most parts of Amhara, Tigray and Benishangul, eastern SNNPR, Gambela, Harari, Dire Dawa, and eastern Oromia. This is because adolescents who are married are exposed to early sexual debut, which results in teenage pregnancy [31]. In most parts of Ethiopia, specifically rural areas, it is common for married girls to become pregnant soon after marriage to prove their fertility. Additionally, family planning utilizations are usually disapproved by the partner, in-laws and the community as a whole as a result of social norms, and little awareness and negative perception also play a role [29,55].

The proportion of being illiterate was significantly positively associated with teenage pregnancy throughout the country according to the MGWR model. This is because literacy increases the understanding of reproductive and sexual health through reading printed materials and contraceptive utilization [26]. Teenagers who are illiterate will have lower self-esteem and want to be accepted and validated by society through early childbearing [56].

Based on the multivariable multilevel logistic regression, literacy, marital status, age of the girls, age of the household head and residence in the Gambela region were significantly associated with teenage pregnancy among adolescents aged 15 to 19 years. Literate teenagers have a

lower risk of pregnancy during adolescence than do illiterate teenagers. This finding is in line with studies conducted in Ethiopia [26], Zambia [57], Philadelphia, the USA [58] and Africa [59]. Giving an emphasis on female literacy will reduce teenage pregnancy. Teenagers who are married are more likely to experience teenage pregnancy than those who are single. This is supported by previously conducted similar studies in Ethiopia [26,31,33], Zambia [57], Uganda [60] and Nigeria [61]. As the frequency of sexual activity is greater among married teenagers, there is a greater likelihood of pregnancy in the absence of contraception than among those who are not married [62].

With respect to age, teenagers aged 17, 18 and 19 years had a greater risk of experiencing teenage pregnancy than did those aged 15 years. This finding is similar to those of studies conducted in Ethiopia [26] and Kenya [63]. This is because as age increases, the probability of engaging in sexual intercourse and marriage, which are proximal determinants of fertility, increases. Consequently, the chances of pregnancy and childbearing will also increase [64]. As the age of the household head increases, the chance of a teenage pregnancy decreases. This finding is consistent with a study from Rwanda [65]. This is because when household heads are older and more mature, they support teens under their household by giving them good parental support, including advising their teen girls on risky sex and fulfilling financial needs [66,67]. In terms of region, the odds of experiencing teenage pregnancy among adolescents aged 15 to 19 years who lived in the Gambela region were greater than those among adolescents in the Tigray region. This is attributable to the region's high unmet need for family planning, greater use of polygamous practices, and early sexual initiation among adolescents, which exacerbates this condition [68].

The key strength of this study was the use of the most recent weighted nationally representative data with a relatively large sample size. To generalize findings at the national level by adjusting the data's hierarchical nature, a multilevel model analysis was performed. The use of spatial and multiscale geographically weighted regression analysis assists in identifying factors that contribute to spatial variation in teenage pregnancy. This research, however, has certain limitations that must be considered when interpreting the findings. Important variables such as age at first sex and marriage and demand for family planning were not included since the mini-EDHS of 2019. In addition, because of the cross-sectional nature of the EDHS data, the cause–effect relationship between the dependent and independent variables cannot be shown.

## Conclusion

There is significant spatial variation in teenage pregnancy among adolescents aged 15 to 19 years. The proportion of being illiterate and the proportion of being married were significantly positively associated with teenage pregnancy throughout the country according to the MGWR model. In the multivariable multilevel logistic regression, literacy, marital status, age of the girls, age of the household head and residence in the Gambela region were significantly associated with the outcome variable. To address this issue, stronger actions must be taken to implement the law, raising the minimum age for marriage to 18 years. Teenage marriage, which is the main cause of teenage pregnancy in Ethiopia, should be changed by addressing cultural norms that prioritize maintaining family status in the community.

Priority should also be given to enhancing levels of literacy and sexual and reproductive health education. In addition to increasing girls' enrollment in formal schools, the government of Ethiopia must increase adult education programs for young women outside of school by incorporating reproductive health issues into the program. The population must also become aware of the consequences and adverse outcomes of teenage pregnancy. Generally, policies

and implementations that focus on preventing early marriage, educating teenage girls, and creating awareness of reproductive health issues need to be strengthened.

## Acknowledgments

We would like to acknowledge the measure DHS program for providing the dataset to use the mini-EDHS 2019.

## Author Contributions

**Conceptualization:** Tsion Mulat Tebeje.

**Investigation:** Tsion Mulat Tebeje, Mesfin Abebe, Fantu Mamo Aragaw, Beminate Lemma Seifu, Kusse Urmale Mare, Ever Siyoum Shewarega, Gizaw Sisay, Binyam Tariku Seboka.

**Methodology:** Tsion Mulat Tebeje.

**Software:** Tsion Mulat Tebeje.

**Writing – original draft:** Tsion Mulat Tebeje.

**Writing – review & editing:** Tsion Mulat Tebeje, Mesfin Abebe, Fantu Mamo Aragaw, Beminate Lemma Seifu, Kusse Urmale Mare, Ever Siyoum Shewarega, Gizaw Sisay, Binyam Tariku Seboka.

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
