## [Editor Report · Decision Letter 0]

24 Jun 2024

PONE-D-23-27072A multi-scale geographically weighted regression analysis of teenage pregnancy and associated factors among adolescents aged 15 to 19 in Ethiopia using the 2019 mini- demographic and health surveyPLOS ONE

Dear Dr. Tebeje,

Thank you for submitting your manuscript to PLOS ONE. After careful consideration, we feel that it has merit but does not fully meet PLOS ONE’s publication criteria as it currently stands. Therefore, we invite you to submit a revised version of the manuscript that addresses the points raised during the review process.

We look forward to receiving your revised manuscript.

Kind regards,

Alfredo Luis Fort, M.D., M.Sc., Ph.D.

Academic Editor

PLOS ONE

Journal Requirements:

5.We note that Figures 2,3,4 and 5 in your submission contain [map/satellite] images which may be copyrighted. All PLOS content is published under the Creative Commons Attribution License (CC BY 4.0), which means that the manuscript, images, and Supporting Information files will be freely available online, and any third party is permitted to access, download, copy, distribute, and use these materials in any way, even commercially, with proper attribution. For these reasons, we cannot publish previously copyrighted maps or satellite images created using proprietary data, such as Google software (Google Maps, Street View, and Earth). For more information, see our copyright guidelines: http://journals.plos.org/plosone/s/licenses-and-copyright.

1. You may seek permission from the original copyright holder of Figures 2,3,4 and 5 to publish the content specifically under the CC BY 4.0 license.  

Please upload the completed Content Permission Form or other proof of granted permissions as an ''Other'' file with your submission.

Additional Editor Comments:

I have seen a similar study like this, with the same problems. The topic and method plus results are of extreme importance. However, we cannot invite Reviewers because they will probably reject the manuscript because, like me, they will find lots of errors, misspellings, and inadequate descriptions. Please take the manuscript to someone who can write proper English, and use my first suggestions (in attached file) to continue improving the wording and descriptions for the rest of the file.

Thank you.

We look forward to receiving your revised manuscript.

---

## [Author Response · Author response to Decision Letter 0]

16 Jul 2024

Subject: Responses to academic editor's comments 

Dear Dr. Alfredo Luis Fort, 

Thank you for taking the time to consider our manuscript titled “A multiscale geographically weighted regression analysis of teenage pregnancy and associated factors among adolescents aged 15 to 19 in Ethiopia using the 2019 mini-demographic and health survey” for the Plos One Journal original research article. We appreciate the time and effort you have dedicated to providing valuable feedback on our manuscript. 

We have taken the comments and concerns into account and made every effort to address them. We agree with all the comments and have incorporated the corresponding revisions into the revised manuscript. We believe that our manuscript has been significantly improved as a result of these revisions, and we hope that our revised manuscript is acceptable for publication in the PLOS One journal.

We would like to thank you once again for your consideration of our work and for inviting us to submit the revised manuscript. We look forward to hearing from you. Our detailed, point-by-point responses to the comments are given below. 

Best regards,

Tsion Mulat Tebeje 

School of Public Health, Dilla University, Dilla, Ethiopia. 

Email: yemarina12@gmail.com (corresponding author)

 

Response to journal’s requirements

Response: We prepared the manuscript according to the journal's requirements and double-checked its compliance before submitting our revised manuscript.

2. Consider depositing your raw data in a repository to ensure your work is read, appreciated and cited by the largest possible audience. You’ll also earn an Accessible Data icon on your published paper if you deposit your data in any participating repository (https://plos.org/open-science/open-data/#accessible-data).

Response: Thank you for your suggestion. We have included it in the revised manuscript and highlighted it in the 'Data Availability' section, specifically on lines 417-418, page 25.

Response: This study involved a secondary data analysis using publicly available data from the MEASURE DHS program and ethical approval and participant consent was not required. This is highlighted in the ethical approval and consent to participate section (lines 194-196, page 9). The data files do not contain any names of individuals or household addresses. We obtained approval from the DHS program to use the dataset and have attached the proof of granted permission as an “other” file with our submission. 

Response: Thank you for the suggestion. We have moved the ethics statement to the methods section in the revised manuscript. 

5. We note that Figures 2,3,4 and 5 in your submission contain [map/satellite] images which may be copyrighted. All PLOS content is published under the Creative Commons Attribution License (CC BY 4.0), which means that the manuscript, images, and Supporting Information files will be freely available online, and any third party is permitted to access, download, copy, distribute, and use these materials in any way, even commercially, with proper attribution. For these reasons, we cannot publish previously copyrighted maps or satellite images created using proprietary data, such as Google software (Google Maps, Street View, and Earth).

You may seek permission from the original copyright holder of Figures 2,3,4 and 5 to publish the content specifically under the CC BY 4.0 license. 

Response: We appreciate your concern regarding the ethical issues. However, the figures (2, 3, 4, and 5) mentioned in our manuscript are not copyrighted; rather, they are the results of spatial analysis conducted using ArcGIS and Python software. The DHS geographic information system, which contains shapefiles with coordinates (latitude and longitude), were obtained from the DHS office by explaining the objective of the study through online requests. To obtain these figures, we imported relevant data extracted from the 2019 Ethiopian mini Demographic and Health Survey. We have attached the permission letter obtained from MEASURE DHS to use the EDHS dataset and geographical location as an “other” file labeled 'AuthLetter_188507'. The Ethiopian administrative boundaries shapefile was obtained from the Ethiopian Central Statistical Agency (CSA). The shape file used to construct the figures can be accessed publicly at https://africaopendata.org/dataset/ethiopia-shapefiles. We incorporated and highlighted this information in captions of figures 2, 3, 4, and 5. 

As a result, the maps presented in our study are not copyrighted. Instead, they are the findings of our spatial analysis, which we performed using ArcGIS and Python on shapefiles and projected CSV files of coordinates. This is the exact process that we followed when conducting our study. Therefore, we can confirm that the figures presented in our study represent our spatial analysis results and are not copyrighted.

Edits requested on the submission

1. We note your current data availability statement:

"The data used in this research are publicly available from the measure DHS program

via an online request at http://www.dhsprogram.com"

As the URL presented in your Data Availability Statement and cover letter links to the DHS Program general website, please provide any direct URL(s), accession numbers, instructions, and additional information (DOI, data set title, etc.) required for researchers to access the specific data underlying the results presented in your study.

Response: Thank you for the suggestion. We have edited the data availability statement by providing the direct URL which links to the login or register for datasets. By registering and logging in, anyone can directly access the dataset. 

2. Please clarify if the data used for the maps are copyrighted/what copyright information or restrictions are attached to it. If the data is copyrighted then please send the below permission form to the owners of the data.

Response: Thank you for asking for clarification. The maps included in our study are not copyrighted from somewhere. They were the output of our work created using ArcGIS version 10.7 and Python 3 software based on the maps on the Ethiopian shapefile obtained from the Ethiopian Central Statistical Agency (CSA), which is publicly available on Open-Africa https://africaopendata.org/dataset/ethiopia-shapefiles, and the GPS data obtained from the measure DHS program after submitting a request along with the study's rationale. Therefore, we assure you that the maps presented in this paper are the result of our work, utilizing appropriate analytical methods and procedures. Furthermore, we have acknowledged the source of the data, and we, as authors, have previously published studies containing similar maps. 

Response to editor’s comments

1. I have seen a similar study like this, with the same problems. The topic and method plus results are of extreme importance. However, we cannot invite Reviewers because they will probably reject the manuscript because, like me, they will find lots of errors, misspellings, and inadequate descriptions. Please take the manuscript to someone who can write proper English, and use my first suggestions (in attached file) to continue improving the wording and descriptions for the rest of the file. 

Response: Thank you for giving us your valuable time and for sharing your valuable input. We fully agree with your suggestion. We have thoroughly reviewed existing similar studies and have appropriately cited them. In the final paragraph of the introduction section, we elucidated how our study differs. The revised manuscript has been meticulously edited and proofread to rectify quality issues, such as misspellings, grammatical errors, and unclear sentences.

---

## [Decision Letter · Decision Letter 1]

23 Aug 2024

A multiscale geographically weighted regression analysis of teenage pregnancy and associated factors among adolescents aged 15 to 19 in Ethiopia using the 2019 mini-demographic and health survey

PONE-D-23-27072R1

Dear Dr. Tebeje,

We’re pleased to inform you that your manuscript has been judged scientifically suitable for publication and will be formally accepted for publication once it meets all outstanding technical requirements.

Within one week, you’ll receive an e-mail detailing the required amendments. There will also be a file with suggestions for the manuscript. When these have been addressed, you’ll receive a formal acceptance letter and your manuscript will be scheduled for publication.

Kind regards,

Alfredo Luis Fort, M.D., M.Sc., Ph.D.

Academic Editor

PLOS ONE

Additional Editor Comments (optional):

Reviewers' comments:

Reviewer's Responses to Questions

**Comments to the Author**

1. If the authors have adequately addressed your comments raised in a previous round of review and you feel that this manuscript is now acceptable for publication, you may indicate that here to bypass the “Comments to the Author” section, enter your conflict of interest statement in the “Confidential to Editor” section, and submit your "Accept" recommendation.

Reviewer #1: All comments have been addressed

Reviewer #2: All comments have been addressed

2. Is the manuscript technically sound, and do the data support the conclusions?

Reviewer #1: Yes

Reviewer #2: Yes

3. Has the statistical analysis been performed appropriately and rigorously? 

Reviewer #1: Yes

Reviewer #2: No

4. Have the authors made all data underlying the findings in their manuscript fully available?

Reviewer #1: Yes

Reviewer #2: Yes

5. Is the manuscript presented in an intelligible fashion and written in standard English?

Reviewer #1: Yes

Reviewer #2: Yes

6. Review Comments to the Author

Reviewer #1: The study’s purpose and aim are clear: to explore the spatial factors that contribute to teenage pregnancy in Ethiopia. But beyond just identifying these factors, we’re diving deeper into what they truly mean for the lives of young girls across different regions. We’re examining a range of determinant like socio-economic conditions, access to education, and cultural practices, that influence whether a teenage girl might become pregnant. By understanding these specific factors, we hope to shed light on the unique challenges faced by adolescents in various parts of the country. This approach isn’t just about numbers; it’s about understanding the real-world implications for these young girls and their futures. The study topic is comprehensive and well researched, but to make it resonate with a wider audience, it could benefit from simplifying some of the more technical parts. For instance, when discussing complex methodological terms, consider breaking them down into more relatable language or using analogies that can bridge the gap between expert knowledge and general understanding. This way, readers who might not be familiar with spatial analysis or statistics can still grasp the importance and impact of your work. Making these adjustments will help ensure that your findings are accessible and meaningful to a broader audience. Good job

Reviewer #2: Dear Author,

Thank you for submitting your manuscript for publication. I have some suggestions to enhance the quality and clarity of the study.

Firstly, I noticed that several similar studies have been using the same data, and methods have been published. To strengthen your manuscript, please highlight the new knowledge your study provides.

2. In the background section, you mention the decline in teenage pregnancy rates since 2000. It would be valuable to explore and discuss potential interventions that implemented during this period that may have contributed to this decline (If any). e.g. "The younglives study".

3. Regarding the results section, I found some discrepancies in the categorization of certain variables, including marital status, wealth index and community exposure to media, which do not add up to the total sample size of teenagers (n=2165). Please review and correct these errors.

4. Additionally, I had difficulty understanding the denominator used to calculate region-specif prevalence rates of teenage pregnancy. To ensure clarity, I suggest including the denominator in square brackets within the results, such as [n=XX/XX]. This will ensure transparency and enable readers to accurately follow the calculations.

For example, when I used the formular (no.teens pregnant in region/no.teen population in region X100) I obtained different rates to what is on Table 1. Providing the denominator will solve the problem.

Overall, your study has potential and addressing these concerns will enhance its quality and contribution to the field.

Regards

P. Nongena

7. PLOS authors have the option to publish the peer review history of their article (what does this mean?). If published, this will include your full peer review and any attached files.

Reviewer #1: No

Reviewer #2: **Yes: **Pelisa Nongena

---

## [Editor Report · Acceptance letter]

29 Aug 2024

PONE-D-23-27072R1 

PLOS ONE

Dear Dr. Tebeje, 

I'm pleased to inform you that your manuscript has been deemed suitable for publication in PLOS ONE. Congratulations! Your manuscript is now being handed over to our production team.

Kind regards, 

on behalf of

Dr. Alfredo Luis Fort 

Academic Editor

PLOS ONE